# Differentiated Thyroid Cancer with Biochemical Incomplete Response: Clinico-Pathological Characteristics and Long Term Disease Outcomes

**DOI:** 10.3390/cancers13215422

**Published:** 2021-10-29

**Authors:** Miriam Steinschneider, Jacob Pitaro, Shlomit Koren, Yuval Mizrakli, Carlos Benbassat, Limor Muallem Kalmovich

**Affiliations:** 1Endocrine Institute, Shamir (Assaf Harofeh) Medical Center, Zerifin 7030000, Israel; miriamsh@shamir.gov.il (M.S.); shlomitks@shamir.gov.il (S.K.); benbassatc@shamir.gov.il (C.B.); 2Sackler Faculty of Medicine, Tel-Aviv University, Tel-Aviv 6997801, Israel; pitaroj@shamir.gov.il (J.P.); YuvalMi@shamir.gov.il (Y.M.); 3Department of Otolaryngology-Head and Neck Surgery, Shamir (Assaf Harofeh) Medical Center, Zerifin 7030000, Israel

**Keywords:** papillary thyroid cancer, biochemical incomplete response, thyroglobulin, prognosis, survival

## Abstract

**Simple Summary:**

Biochemical incomplete response (BIR) is defined as elevated thyroglobulin or rising thyroglobulin antibodies following treatment without structural evidence of disease at 1–2 years after initial treatment. The long-term outcome of such patients is still poorly characterized, with some progressing to structural disease, while others maintain BIR for decades or revert to non-evidence of disease (NED). In this study, we aimed to identify factors that could predict any of the above possible outcomes. In our cohort of 83 BIR patients with a mean follow-up of 12 years, 41% progressed to structural disease. Of them, 11.8% remained BIR, and 38.2% reverted to NED.

**Abstract:**

Although most patients with differentiated thyroid cancer (DTC) and biochemical incomplete response (BIR) follow a good clinical outcome, progression to structural disease may occur in 8–17% of patients. We aimed to identify factors that could predict the long-term outcomes of BIR patients. To this end, we conducted a retrospective review study of 1049 charts from our Differential Thyroid Cancer registry of patients who were initially treated with total thyroidectomy between 1962 and 2019. BIR was defined as suppressed thyroglobulin (Tg) > 1 ng/mL, stimulated Tg > 10 ng/mL or rising anti-Tg antibodies, who did not have structural evidence of disease, and who were assessed 12–24 months after initial treatment. We found 83 patients (7.9%) matching the definition of BIR. During a mean follow-up of 12 ± 6.6 years, 49 (59%) patients remained in a state of BIR or reverted to no evidence of disease, while 34 (41%) progressed to structural disease. At the last follow-up, three cases (3.6%) were recorded as disease-related death. The American Thyroid Association (ATA) Initial Risk Stratification system and/or AJCC/TNM (8th ed.) staging system at diagnosis predicted the shift from BIR to structural disease, irrespective of their postoperative Tg levels. We conclude that albeit 41% of BIR patients may shift to structural disease, and most have a rather indolent disease. Specific new individual data enable the Response to Therapy reclassification to become a dynamic system to allow for the better management of BIR patients in the long term.

## 1. Introduction

The initial assessment of patients with differentiated thyroid cancer (DTC) is based on AJCC/TNM staging for predicting disease-specific mortality and the American Thyroid Association (ATA) risk stratification criteria to predict the risk of persistent or recurrent disease—both are static risk estimates that are obtained immediately after primary therapy [1]. Dynamic risk stratification (DRS) of DTC patients that takes into consideration the response to initial treatment being reassessed at 1–2 years was introduced by Tuttle et al. [2] and revealed significant shifts in the risk categories of DTC patients. Not without a word of caution, this system was adopted by the ATA 2015 guidelines [1], as it does provide a more individualized approach to medical management and a better approach for the prediction of disease recurrence. Response to treatment is described by four categories, including excellent response (ER), biochemical incomplete response (BIR), structural incomplete response (SIR), and intermediate response (IR). These categories are based on post-treatment Tg and anti-Tg levels during follow-up and on imaging studies performed during follow-up. The BIR category is defined as persistent abnormal suppressed and/or stimulated Tg values or elevated anti-Tg antibodies (TgAb) without evidence of structural disease [2]. The BIR group is not uncommon and is seen in 11–22% of DTC patients [1], and while up to 17% of them might develop structural disease, 56–68% will have no evidence of disease (NED) at the last follow up [3]. In this study we aimed to evaluate factors associated with long-term clinical outcome and predictors of structural recurrence in BIR patients.

## 2. Materials and Methods

### 2.1. Subjects and Study Protocol

Patients with DTC who had undergone total thyroidectomy at the department of Otolaryngology-Head and Neck Surgery or its affiliated clinics between 1962 and 2019 (91.4% after 1990) were included. We evaluated the ATA adopted response to treatment criteria in 1049 patients from our local DTC registry who had enough available data for analysis. A total of 795 patients (75.8%) had ER, 91 (8.7%) had BIR, 139 (13.2%) had SIR, and 24 (2.3%) had IR. We defined BIR according to the ATA criteria: sTg > 1 ng/mL, sTg > 10 ng/mL, or rising TgAb in the absence of any evidence of structural disease when assessed at 1–2 years after initial treatment [1]. The clinical features, medical management, and disease outcome of 83 BIR patients were evaluated throughout a mean follow up period of 12 ± 6.6 years. BIR/cure and BIR shift to SIR (BIR-to-SIR) groups were compared for demographic and clinicopathological data, risk factors, medical history, extent of disease, primary treatment, additional treatments, and disease outcome. The following variables were collected: age, sex, pathology (histologic type or subtype: PTC, FTC, Hurtle cell, Tall cell variant, Sclerosing variant, and Insular variant of PTC), TNM stage, ATA risk stratification, laboratory tests, surgery, focality, laterality, lymphocytic infiltration, extrathyroidal extension (ETE), I^131^ treatment, metastases, response to initial treatment, persistent/recurrent disease, disease-free progression, and overall survival. Multifocal disease was defined as more than one tumor foci in one or both lobes. Persistent disease was defined as any evidence of disease based on sonographic and cytology findings, Tg levels, and iodine uptake beyond the thyroid bed in a diagnostic whole-body scan (DxWBS) assessed at 1–2 years after primary treatment. Patients with undetectable stimulated stTg levels upon withdrawal or after recombinant human Thyrotropin (rhTSH), normal findings on neck ultrasound, and a negative DxWBS were defined as disease-free. Recurrent disease was considered in patients who developed any evidence of disease after achieving an excellent response to initial treatment. The 8th edition of the AAJC/TNM system was used for staging. Serum TSH, free T4, free T3, Tg, and TgAb were measured by means of a chemiluminescence assay (Immulite, 2000; Diagnostic Products Corporation, Los Angeles, CA, USA). The functional and analytical sensitivities were 0.9 ng/mL and 0.2 ng/mL, respectively.

### 2.2. Statistical Methods

Categorical variables were presented as numbers and percentages and normally distributed continuous variables as mean and standard deviation, with skewed data being presented as median and the Inter-Quartile range. Data normality was assessed using normal plot and the Shapiro–Wilk test. For differences between groups, we used the independent sample t-test or the Mann–Whitney U test and the chi-square or Fisher’s exact test for continuous and categorical variables as appropriate. Differences on survival and time to recurrence were analyzed using the log-rank test or Cox regression. Since the progression to structural disease was the main outcome of the study, for statistical analysis, patients with persistent BIR and those who reverted to ER were merged as a single group. A *p*-value of less than 0.05 was considered significant. All computations were performed using SPSS version 24 (IBM Corp., Armonk, NY, USA).

## 3. Results

Eighty-three patients with BIR were included in our study, all of whom had undergone total thyroidectomy and 96.4% of whom had received I^131^ after surgery. At the last follow-up, 47 patients (56.6%) achieved ER, 19 (22.9%) remained BIR, and out of the 34 patients who had progressed to SIR (BIR-to-SIR), 13 reverted to ER, 4 to reverted BIR, and only 17 remained SIR after additional treatment. Table 1 shows the clinical characteristics of the BIR patients and the comparison between the BIR and SIR-to-BIR groups, where no significant differences were found.

Data on the tumor characteristics of BIR patients and their primary treatment are shown in Table 2. Neck dissection was performed during total thyroidectomy in 50.6%, of patients, with no differences being found between the BIR-to-SIR and BIR/cure groups. Additionally, there were no significant differences in terms of the histopathological subtypes or the rates of multifocal disease, extrathyroidal extension, and vascular invasion. However, the BIR-to-SIR group presented with statistically significantly bigger primary tumors (*p* = 0.003) and higher T stages (*p* = 0.007) and a nearly significant higher rate of lymph node (LN) involvement (*p* = 0.066). Altogether, the BIR-to-SIR group presented with a significantly higher TNM stage and ATA risk category compared to the BIR patients who did not shift to SIR (*p* < 0.001). Comparing PTC to the FTC BIR patients, no significant differences were found in terms of the clinico-pathological parameters.

For BIR patients, stimulated post-op and last visit Tg levels were 12.3 ± 14 and 14.4 ± 28 ng/mL, respectively; suppressed post-op and last visit Tg levels were 3.9 ± 11.4 and 4.7 ± 15 ng/mL, respectively (*p* = ns). The comparison of Tg levels between the BIR and BIR-to-SIR groups showed non-significant differences, with post-op and last visit stTg levels of 8.7 ± 8 vs. 19.6 ± 20 ng/mL and 9.1 ± 15 vs. 27 ± 46 ng/mL and post-op and last visit-suppressed Tg levels of 4.7 ± 13 vs. 1.8 ± 1.8 ng/mL and 5.1 ± 17 vs. 3.9 ± 7.3 ng/mL, respectively (*p* = ns). Even more so, a comparison of the Tg levels at the last follow-up of the BIR-to-SIR patients who reverted to ER with those who remained SIR showed non-significant differences, with sTg levels of 0.04 ± 0.09 vs. 13.7 ± 7.5 and stTg levels of 1.0 ± 6.6 vs. 78.5 ± 51.6 (*p* = ns).

Table 3 shows the additional treatments and disease outcomes observed during follow-up. Post operative I^131^ was given to nearly all BIR patients (96.4%). As expected from their higher ATA risk score, the initial I^131^ dose was higher for the BIR-to-SIR patients (*p* = 0.028). While no recurrences were recorded for the BIR patients, structural recurrence in the BIR-to-SIR group was local in 50% of the patients, LN was found in in 26.5% of patients, and distant metastases (DM) were found in 42.4% of patients, which resulted in re-operation (44.1%), additional I^131^ therapy (97.1%), and external beam radiotherapy (23.5%). Interestingly, driven by the persistent thyroglobulin levels, additional I^131^ therapy was given to 58.7% of BIR patients as well. Nevertheless, in the BIR-to-SIR group, the cumulative I^131^ dose was 350 mCi compared to 180 mCi in the BIR group (*p* = 0.001).

At the end of the study, the all-cause mortality rate was 10.8%, while disease-related mortality was 3.6%, both of which were entirely in the BIR-to-SIR group (3/34, 8.8%). All three patients had distant metastases to the lungs, and one of them had a Tall Cell variant of PTC. Overall disease-free survival is shown in Figure 1.

## 4. Discussion

Biochemical incomplete response to treatment at the first two years postoperatively does not always imply that the patient has a persistent thyroid cancer. In some cases, an incomplete thyroidectomy results in persistent low level Tg values arising from non-malignant thyrocytes. To identify those patients with true BIR, clinicians need to integrate the primary static assessment with prognostic factors during follow-up, which will be discussed here. BIR is observed among 11–19% of low-risk, 21–22% of intermediate-risk, and 16–18% of high-risk patients as per the ATA [2]. In newer publications, BIR is observed at about 15–20% in all risk categories, with decreasing ER and increasing SIR according to risk category [4,5]. We analyzed 83 patients from a cohort of 1049 (7.9%) with a biochemical incomplete response throughout a mean follow-up of 12.0 ± 6.6 years. During the follow-up 59.0% reverted to NED or remained BIR, while 41.0% progressed to structural disease.

### 4.1. Tg Levels and Timing Definition

There are variations in both the time interval of TG level testing and in the cutoff levels between ER and BIR. According to the ATA, BIR is defined as negative imaging and suppressed Tg ≥ 1 ng/mL or Stimulated Tg ≥ 10 ng/mL or rising anti-Tg antibody levels. While the specific cutoff levels of Tg that optimally distinguish normal residual thyroid tissue from persistent thyroid cancer have not been determined, rising Tg values over time are suspicious for growing thyroid tissue or cancer. Most authors regard Tg < 1 ng/mL as ER, while others refer to stimulated Tg < 2 mg/mL as the cutoff between ER and BIR [6]; some use stimulated Tg [7,8,9], and others use suppressed or trends in anti Tg Ab titers. The time interval for Tg level testing varies, potentially starting from 8–12 months after initial therapy [10] to 2 years post initial therapy [2]. The response-to-treatment assessment interval does also differ between studies, ranging from 6 to 24 months after initial therapy [1]. Our group measured Tg levels at 1 year postoperatively, and the last visit was adopted as the ATA cutoff point. Nevertheless, we did not find a significant difference in the Tg levels (either stimulated or suppressed) among the patients who developed structural recurrence compared to patients who remained BIR or who reverted to NED. Of note, only a small subset of patients had enough serial Tg measurements available to address the doubling time as a possible predictor [11,12,13]. Applying dynamic risk stratification based on post treatment Tg levels only revealed that a minority of BIR patients will eventually develop a structural disease, and the Tg trend was found to be useful in identifying those patients [14,15]. Nevertheless, additional treatments do not invariably achieve a biochemical cure, which is partly due to highly sensitive laboratory tests [3,9,16,17], nor do additional treatments prevent the shift from BIR to SIR.

### 4.2. Age as a Prognostic Factor for Structural Recurrence

Response to treatment is associated with patient age: according to the ATA, the ER rate among high-risk patients is significantly higher among younger (<55 year) patients, while SIR is significantly higher among older patients [1]. Shah et al. [18] showed that high-risk (HR) patients who are younger than 55 years of age at diagnosis will have twice the chance of achieving an excellent response when compared to older patients. Kim et al. [19] found more gene mutations in patients older than 55 years of age at diagnosis, which was associated with more aggressive pathways compared to younger patients. In line with previous reports [20,21,22], our data show that age does not predict the BIR-to-SIR shift.

### 4.3. Tumor Characteristics

The only significant differences between BIR/cure and BIR-to-SIR patients were the ATA risk and advanced TNM stage at the time of diagnosis.

Most of our BIR patients (72.8%) were at a low risk (LR) as per the ATA, while only 8.4% were high risk (HR). However, the ATA risk in the BIR-to-SIR group was 55.9% LR and in 14.7% HR compared to 85.1% and 4.3% in the BIR/cure group (*p* < 0.001). Thus, initial ATA risk assessment was a significant predictor for structural recurrence in BIR patients. At diagnosis, 13/83 patients (16.1%) were at T-stage II/III; 26.4% were BIR-to-SIR patients, and 8.5% were in the BIR-to-cure group (*p* = 0.001). Thus, an advanced tumor stage at diagnosis appears to be another prognostic factor for structural recurrence in BIR patients. Similar findings were reported by Vaisman et al. [23], who also reported poor outcomes among HR patients (68% with persistent/recurrent disease and 16.5% of death) compared to intermediate-risk (35.7% of persistence and 0.9% of death) and low-risk patients (12.5% of persistence and 0% of death, *p* < 0.001). In his study, only 15.5% of the HR patients achieved NED at the last follow-up, as opposed to 87.5% of patients in the LR group (*p* < 0.001). We found a disease-related mortality of 3.6%, which was predominantly in the high-risk ATA stage and in the BIR to SIR group; this is in line with others [24,25], demonstrating that the mortality and progression of DTC mostly occur in the structurally incomplete status.

### 4.4. Developing Structural Recurrence

In our cohort, 41% progressed to structural recurrence, with 50% of them remaining SIR at last visit. In Vaisman et al., 55.6% of the BIR patients had NED at the end of follow up. Although patients with BIR generally have a favorable outcome, some authors [6,26,27] have found immunohistochemistry for BRAFV600E positivity as a risk factor for structural recurrence. Zern et al. [6] found that 34.4% of BIR patients developed a structural recurrence and that all of these patients were associated with BRAFV600E positive tumors. On the other hand, Ito et al. [28] found that patients with BRAFV600E were equally distributed amongst both high- and low-risk cases. So far, no single marker can predict clinical outcome or structural recurrence [29], and thorough disease surveillance is required for BIR patients [25,30].

### 4.5. Limitations

Our study has several limitations. Patient data were collected from different medical centers, as some had therapy completed outside of our center; consequently, some data were missing. To fully assess the long-term risk of recurrence, patients need to have a standardized follow-up for many years; however, some of our patients were lost to follow up, which is not uncommon after achieving ER, and were relocated to another hospital or had their follow-up continued in community clinics. The lack of serial Tg measurements precluded the use of Tg slope in the prediction analysis. Regarding histological definitions, new classifications were not applied in our cases and could have been included as differentiated thyroid cancers, such as non-invasive follicular thyroid neoplasm with papillary-like nuclear features (NIFTP) or papillary tumor capsule integrity [31]. On the other hand, the strengths of our study were a large cohort of 1049 patients with a long follow up of 12 years.

## 5. Conclusions

Well-differentiated thyroid cancer patient care has evolved from having an identical treatment and follow-up for all patients to an individualized treatment approach based on dynamic risk stratification. Although many BIR patients will eventually shift to NED, some will progress to structural disease; therefore, they should be followed according to their risk of recurrence (RR) and their serial Tg measurement trends, with varying degree of suspicion based on their initial ATA risk stratification and TNM stage.

## Figures and Tables

**Figure 1 cancers-13-05422-f001:**
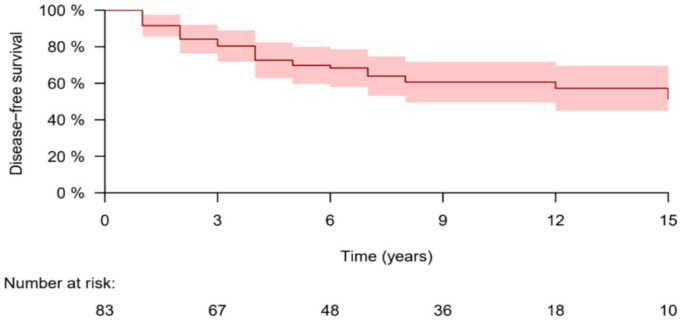
Overall disease-free survival for biochemically persistent DTC patients.

**Table 1 cancers-13-05422-t001:** Patient demographics and clinical characteristics.

Variable	All	BIR-to-SIR	BIR/Cure	*p* Value
*n*	83	34	49	
Age, mean ± SD	47.0 ± 16.4	48.3 ± 16.7	46.2 ± 16.2	0.237
Female, sex *n* (*%*)	58 (69.9%)	22 (64.7%)	36 (73.5%)	0.355
Thyroid function pre-op, *n* (*%*)				0.650
Euthyroid	72 (87.8%)	30 (90.9%)	42 (85.7%)	
Hypothyroidism	5 (6.1%)	2 (6.1%)	3 (6.1%)	
Hyperthyroidism	5 (6.1%)	1 (3.0%)	4 (8.2%)	
Hashimoto thyroiditis, *n* (*%*)	27 (33.3%)	10 (31.3%)	17 (34.7%)	0.658
Familial history, *n* (*%*)	6 (7.2%)	3 (8.8%)	3 (6.1%)	0.574
Past radiation exposure, *n* (*%*)	8 (9.6%)	5 (14.7%)	3 (6.1%)	0.157

SIR, structural incomplete response; BIR, biochemical incomplete response.

**Table 2 cancers-13-05422-t002:** Tumor characteristics.

Variable	All	BIR-to-SIR	BIR/Cure	*p* Value
*n*	83	34	49	
Neck dissection, *n* (*%*)	42 (50.6%)	17 (50.0%)	25 (51.0%)	0.593
Post-op I^131^ treatment, *n* (*%*)	80 (96.4%)	34 (100%)	46 (93.9%)	0.359
I^131^ initial dose, median (mCi)	150 (80–150)	150 (100–150)	100 (65–150)	0.028
Pathological subtype, *n* (*%*)				0.474
PTC	64 (77.1%)	25 (73.5%)	39 (79.6%)	
FTC	14 (16.9%)	6 (17.6%)	8 (16.3%)	
Other *	5 (6.0%)	3 (8.8%)	2 (4.1%)	
Multifocal, *n* (*%*)	49 (59.0%)	21 (61.8%)	28 (57.1%)	0.971
Bilateral, *n* (*%*)	32 (39.5%)	15 (45.5%)	17 (35.4%)	0.585
Tumor size, median (mm)	18 (12,28)	20 (14,40)	15 (11,24)	0.003
T stage, *n* (*%*)				0.017
1	51 (63.0%)	18 (52.9%)	33 (70.2%)	
2	16 (19.8%)	6 (17.6%)	10 (21.3%)	
3	11 (13.6%)	8 (23.5%)	3 (6.4%)	
4	3 (3.7%)	2 (5.9%)	1 (2.1%)	
Extrathyroidal extension, *n* (*%*)	18 (21.7%)	10 (29.4%)	8 (16.3%)	0.201
Muscle extension *n* (*%*)	7 (8.9%)	4 (11.8%)	3 (6.7%)	0.385
Blood vessel involvement *n* (*%*)	7 (8.5%)	4 (11.8%)	3 (6.3%)	0.404
LN involvement, *n* (*%*)	36 (43.4%)	18 (52.9%)	18 (36.7%)	0.066
Extra LN extension, *n* (*%*)	1 (1.2%)	0 (0%)	1 (2.0%)	0.541
TNM stage, *n* (*%*)				0.001
1	68 (84.0%)	25 (73.5%)	43 (91.5%)	
2	11 (13.6%)	8 (23.5%)	3 (6.4%)	
3	2 (2.5%)	1 (2.9%)	1 (2.1%)	
ATA risk assessment, *n* (*%*)				<0.001
Low	59 (72.8%)	19 (55.9%)	40 (85.1%)	
Intermediate	15 (18.5%)	10 (29.4%)	5 (10.6%)	
High	7 (8.6%)	5 (14.7%)	2 (4.3%)	

SIR, structural incomplete response; BIR, biochemical incomplete response; PTC, papillary thyroid cancer; FTC, follicular thyroid cancer; ATA, American Thyroid Association; LN, lymph nodes. * Other—PTC variants including Insular, Tall cell, Sclerosing; Hurtle cell carcinoma.

**Table 3 cancers-13-05422-t003:** Additional treatment and follow-up.

Variable	All	BIR-to-SIR	BIR/Cure	*p* Value
*n*	83	34	49	
Local recurrence, *n* (*%*)	17 (20.5%)	17 (50.0%)	-	-
LN recurrence, *n* (*%*)	9 (10.8%)	9 (26.5%)	-	-
Distant metastases, *n* (*%*)	14 (16.9%)	14 (42.4%)	-	-
Re-operation, *n* (*%*)	16 (19.3%)	15 (44.1%)	-	-
Additional I^131^, *n* (*%*)	60 (75.0%)	33 (97.1%)	27 (58.7%)	0.001
I^131^ cumulative dose, median (mCi)	288 (162.5350)	350 (300,450)	180 (150,300)	<0.001
External beam radiotherapy, *n* (*%*)	9 (11.0%)	8 (23.5%)	1 (2.1%)	<0.001
Death, *n* (*%*)	9 (10.8%)	7 (20.6%)	2 (4.1%)	0.180
Disease-related deaths *n* (*%*)	3 (3.6%)	3 (8.8%)	0 (0%)	0.104

LN, lymph node.

## Data Availability

The data presented in this study are available on request from the corresponding author. The data are not publicly available due to institutional review board restrictions.

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
