# Peer review of "Differentiated Thyroid Cancer with Biochemical Incomplete Response: Clinico-Pathological Characteristics and Long Term Disease Outcomes"

_cancers, 2021, doi:10.3390/cancers13215422_

Round 1
Reviewer 1 Report
This study aimed at evaluating the possible predicting factors of patients with biochemical incomplete response after total thyroidectomy for disease progression and long-term outcomes. Overall, the manuscript is well written and of interest. However, some specific comments should be addressed before the paper can be considered for publication.
Introduction
This section is rather poor. The authors should implement the background with additional data and statistics.
Abstract and Methods.
The authors stated the all patients had PTC but Table 2 reports that some of patients with BIR had FTC and other subtypes of thyroid cancer. Could you explain this point?
Results
Table 4. Please correct the typo in “suppressed”.
Discussion
Paragraph 4.1
At the end of the paragraph please correct the typos in “stimulated”, “suppressed” and “achieve”.
Paragraph 4.2
Please use the lowercase initial in “the” and “cure” and amend the typo in “showed”, “compared”, and “without”.
Please provide the full name of “HR”.
Paragraph 4.3
Please provide the full name of “LR”.
What data does the last sentence of the paragraph refer to?
Paragraph 4.4
Please check the grammar of the last sentence and rephrase accordingly.
Conclusions
Please provide the full name of WDTC and RR.
Author Response
Thank you very much for your evaluation. We hereby enclose our changes accordingly.
Introduction: The text was re edited accordingly.
Abstract and Methods: We corrected to “Differentiated thyroid cancer” as we had 16% FTC in the study population.
Results: typo corrected. Table 4 was deleted and data is included in the text.
Discussion
4.1- Thank you, typo corrected
4.2- Typo corrected and full name provided thank you
4.3- full name provided
The paragraph was re edited and clarified - T stage and risk stratification aid in predicting structural recurrence.
4.4- Paragraph was edited.
Conclusions - full names were provided.
Reviewer 2 Report
This is an interesting observational study aiming at investigating possible predicting factors of progression to structural disease in patients who underwent total thyroidectomy for papillary thyroid cancer (PTC) with biochemical incomplete response (BIR). The study is not conclusive because of some limitations that the Authors clearly identified but outlines a certain patient tailored follow-up based on potential predicting factors. Some minor criticisms should be straightened before considering the study suitable for publication. The acronym ATA should be explained in the abstract. To avoid repetition all raw data shown in the Tables should be eliminated in the text. Table 4 is not cited in the Results and can be eliminated or simply grouped in one sentence in the results as no statistical significance was reach in any of the variables. All significant data shown in the results should be emphasized in the conclusions notwithstanding well known limitations.
Author Response
Thank you for your remarks. Here are the corrections we made accordingly:
ATA - explained
Table 4 was eliminated and grouped as text sentence in the results.
We emphasized data in the discussion (paragraph 4.3) and conclusions.
Reviewer 3 Report
The authors tried to draw factors to recurrent diseases in PTC patients with BIR.
Although this study was designed with long-term follow-up, study methods and results are well known. So, I could not find any originality or clinical meaning in your study.
Moreover, some definition was wrong (e.g., SIR in last visit?) and several methods might be ignored to understand the results (including Tg cutoff according to TSH stimulation.
When considering these factors, this study is insufficient to include 'advances in thyroid cancer'.
Author Response
Thank you for revising our study.
Regarding the comment about SIR in last visit - we further clarified that among 40% patients that progressed from BIR to SIR, 11.8% remained BIR in the last visit, 38.2% reverted to NED and 50% remained SIR after additional treatment.
Round 2
Reviewer 1 Report
The authors addressed my previous comments and the manuscript is suitable for publicazione in present form.
Author Response
We would like to thank you again for your review and constructive comments.